# Damage Localization on Composite Structures Based on the Delay-and-Sum Algorithm Using Simulation and Experimental Methods

**DOI:** 10.3390/s23094368

**Published:** 2023-04-28

**Authors:** Cedric Bertolt Nzouatchoua, Mourad Bentahar, Silvio Montresor, Nicolas Colin, Vincent Le Cam, Camille Trottier, Nicolas Terrien

**Affiliations:** 1Laboratoire d’Acoustique de l’Université du Mans (LAUM), UMR CNRS 6613, Institut d’Acoustique-Graduate School (IA-GS), CNRS, Le Mans Université, 72085 Le Mans, France; 2IRT Jules Verne, NANTES Université, F-44000 Nantes, France; 3Laboratoire Structure et Instrumentation Intégrée (SII), Inference for Structures (I4S) Team, Département Composants et Systèmes (COSYS), Université Gustave Eiffel, 44344 Bouguenais Cedex, France

**Keywords:** structural health monitoring, PZT network, Lamb waves, anisotropic composite structures, impact damage, delay-and-sum

## Abstract

Damage detection and localization based on ultrasonic guided waves revealed to be promising for structural health monitoring and nondestructive testing. However, the use of a piezoelectric sensor’s network to locate and image damaged areas in composite structures requires a number of precautions including the consideration of anisotropy and baseline signals. The lack of information related to these two parameters drastically deteriorates the imaging performance of numerous signal processing methods. To avoid such deterioration, the present contribution proposes different methods to build baseline signals in different types of composites. Baseline signals are first constructed from a numerical simulation model using the previously determined elasticity tensor of the structure. Since the latter tensor is not always easy to obtain especially in the case of anisotropic materials, a second PZT network is used in order to obtain signals related to Lamb waves propagating in different directions. Waveforms are then translated according to a simplified theoretical propagation model of Lamb waves in homogeneous structures. The application of the different methods on transversely isotropic, unidirectional and quasi-transversely isotropic composites allows to have satisfactory images that well represent the damaged areas with the help of the delay-and-sum algorithm.

## 1. Introduction

Maintenance operations are periodically carried out in industrial environments using non-destructive testing (NDT) techniques to ensure the reliability of structures. However, these techniques often have several disadvantages, such as the immobilization of the structure throughout the inspection, the need for qualified operators and long inspection times, depending on the dimensions to be inspected. Structural Heath Monitoring (SHM) techniques have therefore been proposed to develop autonomous systems for continuous monitoring, inspection and damage detection of structures with minimal manpower involvement [1]. Two types of methods can be distinguished in the practice of SHM. Passive methods measure the response of the structure continuously and monitor their evolution with respect to the given thresholds; active methods, in addition to sensors, use actuators that generate stress waves to interrogate the structure. Among active methods, the ones based on the use of guided ultrasonic waves have shown a good sensitivity to the presence of damage [2]. In the case of plate-like structures, Lamb waves can be generated in symmetric (S) and anti-symmetric (A) modes and can propagate over long distances with low attenuation [3]. In general, the fundamental Lamb modes A0 and S0 are used for practical reasons often related to the interpretation of signals. In particular, the S0 mode has a higher sensitivity to structural thickness damage such as delamination, while the A0 mode has a higher sensitivity to surface damage such as cracking or corrosion [4]. In SHM, Lamb waves’ inspection is based on the use of several sensors, each of which plays the role of transmitter and receiver in an alternating way. Lamb waves can interact in different ways (reflection, scattering, conversion) with the existing damage. Signal processing is then performed to the collected signals in order to extract different parameters such as time of arrival, amplitude, energy, etc. Different imaging algorithms can be used to detect and locate the damage within the inspection area. In general, PZT sensors are very often used in Lamb wave inspections and can be tuned to generate only one Lamb mode of interest [5,6].

There are several works in the literature dealing with Lamb wave damage localization using PZT arrays on aluminium and composite plates [7,8,9,10,11]. In general, these works are based on the extraction of a specific Damage Index (DI) from the detected signals. Several algorithms can therefore be implemented to perform the localization of the different damages by imaging the inspection area delimited by the PZT sensors. Some algorithms are based on the monitoring of the inspection area using the same PZT network. This can be performed by first acquiring baseline signals related to the initial state in which the structure is healthy. Signals related to the subsequent state of the structure, possibly a damaged state, are compared to baseline signals, where a significant change above a previously defined threshold is considered as a sign of an existing anomaly [12]. Residual signals (damage scattered signals), corresponding to the difference between damaged and baseline signals, are then used in the imaging algorithms. Lamb wave tomography [13,14,15,16] is a good way to quantitatively evaluate damage imaging reconstruction but requires a high number of transducers. A simple example is the reconstruction algorithm based on the probabilistic damage inspection called RAPID. The latter uses the correlation between signals taken at the healthy and damaged states, which is quantified statistically by the Signal Difference Coefficient (SDC) [17]. Despite the need for a large number of transducers, the algorithm does not require a priori knowledge of the dispersion properties of the involved Lamb waves. For the RAPID algorithm, the shape of the PZT array distribution has a great influence on the imaging quality. Indeed, a good PZT distribution is the one that allows to have a homogeneous actuator–sensor path flow in the inspection area, thus avoiding focusing points that can create artifacts in the final image. Note that a circular shape of the PZT array can provide optimal results [18]. A normalization factor can be introduced to compensate the effect of the shape of the PZT network [19,20]. Hua et al. [21] have proposed an optimization of the RAPID algorithm by transforming the SDC into LSDC (Local Signal Difference Coefficient), a new parameter that uses optimal signal lengths. On the other hand, algorithms based on the sparse reconstruction, i.e., exploiting the sparse nature of damages, are also used based on the propagation of Lamb waves. These algorithms attempt to find a small set of locations that might contain damage consistent with the residual signals [22,23]. However, they face difficulties in analytically building a dictionary. The delay-and-sum (DAS) algorithm, adopted from the radar community, is based on the exploitation of the residual signals and the group velocity of the propagating Lamb modes. The residual signals from all transmitter–receiver paths are shifted according to an appropriate time-shift rule and then summed up to give an average signal [24,25]. The delay-and-sum algorithm offers the possibility of locating several damages, even in large and complex structures, by taking in account multi-damage strategies such as the one presented in [26,27] based on extensions of the delay-and-sum algorithm.

Baseline algorithms are often criticized for not taking into account environmental conditions such as temperature during the acquisition of Lamb waves signals at different states. Despite the existence of temperature compensation methods [28], one should note that it is not always possible to have a state at which the structure is healthy, especially on operational structures. In this case, the current trend is to develop methods that do not need the pristine state, where only measurements in the current state are considered. Methods using time reversal [29,30] are considered as baseline-free. They are based on the spatial reciprocity and time invariance of linear wave equations. Other baseline-free methods are based on the baseline algorithm, with the exception that the reference state is constructed by exploiting measurements of the current or damaged state. Salmanpour et al. [31] have proposed a method in which the baseline state is constructed simultaneously with the damaged state by dividing the inspection region into symmetrical areas; in the absence of a pristine state, baseline signals are obtained by mapping transducers from a baseline area to localize damage within an interrogation area, where the size of the baseline area is comparable to that of the interrogation area. In [32,33], baseline-free methods ERAPID and nonlinear RAPID, inspired by the baseline algorithm RAPID, are described.

The novelty of the present work is in its use for the generation of baseline signals, a small healthy area (i.e., a spatial reduction of the main network) which considers the experimentally determined anisotropy of the composites studied, combined with an analytical construction based on a simple theoretical propagation model of the elastic guided waves.

The paper presents different methods that allow for obtaining baseline signals. Section 2 presents the damage localization methods based on the delay-and-sum algorithm in the case of anisotropic materials for simulated and experimental results. Section 3 presents the study of a transverse isotropic composite plate taken at the damaged state (impacted state), where impact has been modeled as conical in shape with degraded mechanical properties (details of this work can be found in [34]). Baseline signals are obtained based on 3D numerical simulation results performed using the SHM module of the CIVA software. In Section 4 and Section 5, results related to the characterization of a curved wind turbine blade sample and a carbon fiber composite plate are presented, respectively. For these two samples, the elastic properties, draping and material symmetry were not previously known. This paper presents the experimental characterization of anisotropy related to both samples as well as residual signals obtained to image damaged areas with the delay-and-sum algorithm. The paper ends with a conclusion and perspectives of the present work.

## 2. Damage Localization Methods Based on Delay-and-Sum Algorithm

This section presents the methods that are applied to structures that were previously damaged, for which no reference or healthy state data are available. The methods are based on the use of PZT network to delineate the inspection area containing damage. Structures are discretized into uniform grids of regularly spaced pixels. Images are obtained by calculating the contrast values at each pixel based on the probability of presence/absence of damage at the different locations. The fundamental Lamb modes A0, S0 and SH0 are used for the reconstruction of the different images. The methods presented below use the delay-and-sum algorithm adjusted to take into account the anisotropy of the composite structures.

### 2.1. Simulation Method

The method presented in Figure 1 is based on the characterization of a damaged composite plate for which the pristine state is not available. By identifying the elasticity tensor elements, baseline signals (i.e. for the healthy structure) related to the propagating Lamb waves were obtained with the help of numerical simulations performed using CIVA software [35,36]. Simulations take into account the real experimental configuration (plate geometry, excitation signal and frequency, number of PZTs, scan configuration, dimensions and positions of the PZT sensors, sampling frequency and simulation time). Then, simulation results are subtracted from the experimentally obtained signals in the damaged state in order to obtain residual signals containing the damage signature. The last step of the proposed method is the application of the delay-and-sum algorithm to image damage in the inspection area.

### 2.2. Experimental Method

The first method uses a numerical simulation model to construct baseline signals in which the excitation parameters are exactly the same as in the experimental determination of damaged signals. However, one should be aware that the computation time can be very long, and the real propagation conditions in terms of “wave/structure” interaction are not always easy to implement. Furthermore, the determination of the elastic constants by inverse method remains a delicate task, despite the existence of various approaches such as the one based on the Lamb waves [37,38]. In composites structures, the sparse nature of damages makes the density of damaged areas low compared to healthy areas [39]. This makes it possible to find a healthy area on the structure and place a second PZT network with the same shape as the main one around the inspection area. In the second method shown in Figure 2, the second PZT network (image of the main PZT network by a dilation as can be seen for square and circular PZT networks in Figure 3a,b) is used to acquire intermediate signals.

Xu et al. describe a theoretical approach to the Lamb waves propagation model in [40]. The latter is applied and adapted to our study in order to obtain the reconstructed baseline signals from intermediate signals, as described below.

It is assumed that a distance *r* (in m) in Figure 3c is the difference between the length *D* (in m) of a path in the main network and *d* (in m), the equivalent in the second network of PZTs. The complex signal *S(t)* of an intermediate signal *I(t)* in the time domain can be written as
(1)$$S\left(t\right)=I\left(t\right)*\exp\left(iw_{c}t\right),$$
with $w_{c}$ (in rad/s) being the angular central frequency of the excitation signal, *t* (in s) is the travel time, and *i* is the imaginary unit. In the frequency domain, the previous equation can be expressed as a function of the angular frequency *w* (in rad/s): (2)$$\hat{S}\left(w\right)=\int_{-\infty }^{+\infty }S\left(t\right)*\exp\left(-iwt\right)dt,$$
leading to: (3)$$\hat{S}\left(w\right)=\int_{-\infty }^{+\infty }I\left(t\right)*\exp[\left(-i\left(w-w_{c}\right)t\right]dt.$$

We can therefore write:(4)$$\hat{S}\left(w\right)=\hat{I}\left(w-w_{c}\right).$$

The frequency signal corresponding to the baseline signal *B(t)* can be expressed as
(5)$$\hat{B}\left(w\right)=\frac{1}{\sqrt{r}}*\hat{S}\left(w\right)*\exp\left(-ik\left(w\right)r\right),$$
where k(w) (in rad/m) is the wavenumber of the corresponding Lamb wave mode. The time representation of Equation (Equation 5) can be written as
(6)$$B\left(t\right)=\frac{1}{\sqrt{r}}*TF^{-1}\left\{\hat{I}\left(w-w_{c}\right)*\exp\left(-ik\left(w\right)r\right)\}\right.$$
where $TF^{-1}$ represents the inverse Fourier transform. Assuming the linearity of the wavenumber as a function of frequency, i.e., in a low dispersive region on the Lamb waves dispersion curves, the wavenumber can be written in the vicinity of $w_{c}$ as
(7)$$k\left(w\right)=\alpha +\beta \left(w-w_{c}\right)+0\left(w-w_{c}\right),$$
with α (in rad/m) and β (in s/m) being expressed by
(8)$$\alpha =k\left(w_{c}\right)=\frac{w_{c}}{V_{p}\left(w_{c}\right)}\;\mathrm{and}\;\beta =k^{\prime }\left(w_{c}\right)=\frac{dk(w)}{dw}|\left(w=w_{c}\right)=\frac{1}{V_{g}\left(w_{c}\right)}$$
where $V_{p}$ (in m/s) and $V_{g}$ (in m/s) represent the phase velocity and group velocity of the Lamb mode under consideration, respectively. The term 0 ($w-w_{c}$) and higher order terms are considered as interference terms and are neglected. Thus, Equation (Equation 6) can now be written as follows: (9)$$B\left(t\right)=\frac{1}{\sqrt{r}}*TF^{-1}\left\{\hat{I}\left(w-w_{c}\right)*\exp\left[-i\left(\alpha +\beta \left(w-w_{c}\right)\right)r\right]\right\},$$
leading to
(10)$$B\left(t\right)=\frac{1}{\sqrt{r}}*exp\left(-i\alpha r\right)*\;exp\left(iw_{c}t\right)*\;I\left(t-\beta r\right).$$

Using Equation (Equation 1) to make *S(t)* appears in the previous equation, it follows that
(11)$$B\left(t\right)=\frac{1}{\sqrt{r}}*exp\left(-i\alpha r\right)*\;exp\left(iw_{c}\beta r\right)*\;S\left(t-\beta r\right).$$
The signal *B(t)* can be finally expressed as
(12)$$B\left(t\right)=\frac{1}{\sqrt{r}}*\left[\exp\left(-i\frac{r*w_{c}}{V_{p}\left(w_{c}\right)}\right)*\exp\left(i\frac{r*w_{c}}{V_{g}\left(w_{c}\right)}\right)\right]*S\left(t-\frac{r}{V_{g}\left(w_{c}\right)}\right).$$

This signal is the product of three terms. The first one represents Lamb waves’ attenuation inversely proportional to the square root of the distance r. The second term (time independent) with exponentials, represents the phase variation; It vanishes when the phase velocity ($V_{p}$) is equal to the group velocity ($V_{g}$). The third term corresponds to a time shift due to propagation.

The size of the second PZT network is controlled by the scale’s value of the dilation (the closer the scale is to 0, the smaller the second network, while a value of 1 indicates a secondary network with a same size as the main network); the placement is determined by the position of the dilation center. These two characteristics are chosen according to the largest healthy subarea available within the main network. In practice, that subarea is not known and a coherent approach to determine it may be to create several non-overlapping second networks by several dilations with the same scale but with different centers in such a way as to occupy the entire inspection area within the main network. An acquisition is then performed for all these second networks with the same scan configuration. The second network with the highest residual signal amplitude compared to the others can be considered to be located in a healthy zone [31]. It will then be considered for further processing. Furthermore, it is probable that for certain paths, baseline signals reconstructed from intermediate signals do not contain all the wave packets corresponding to the reflections on the edges of the structure that the damaged signals have, since the main PZT network is closer to the edges of the structure than the second PZT network. In this case, after baseline subtraction, residual signals will not only contain the damage scattered signal, but also the Lamb wave response of the geometric features. Since this part of the signal is not useful for the damage localization, its presence is not very detrimental and can even be minimized by optimizing the acquisition time of damaged signals. In addition, the delay-and-sum algorithm only uses an envelope amplitude value at a specific time (well determined by the knowledge of the path’s direction velocity and length), in the residual signal, thus circumventing this problem.

### 2.3. Delay-and-Sum Algorithm Applied to Anisotropic Materials

An illustration for a single pair of PZTs between transmitter *i* of coordinates ($x_{i}$,$y_{i}$) and receiver *j* of coordinates ($x_{j}$, $y_{j}$) is given in Figure 4. It can be seen that the corresponding residual signal in time domain $RES_{ij}$ (t) is delayed through the pixel *P* of coordinates ($x_{P}$, $y_{P}$). If the network contains a number $N_{T}$ of PZTs, the same is true for all pairs used in the acquisitions, giving rise to a number $N_{S}$ of signals. Each pixel in the grid undergoes this action and the contrast value at pixel *P* is obtained by the following formula: (13)$$I_{P}\left(x_{P},y_{P}\right)=\frac{1}{N_{s}}\sum_{i=1}^{N_{T}}\sum_{\begin{array}{c} j=1\\ j\#i \end{array}}^{N_{T}}{\mathrm{ERES}}_{ij}\left(t_{ij}\left(x_{P},y_{P},\alpha \right)\right),$$
where $ERES_{ij}$ is the envelope of the residual signal $RES_{ij}$ between transmitter *i* and receiver *j* and is obtained by a Hilbert transform. The time of flight $t_{ij}$ ($x_{P}$, $y_{P}$, α) of the signal along the virtual path *i-P-j* is given by
(14)$$t_{ij}\left(x_{P},y_{p},\alpha \right)=\frac{d_{i-P}}{V_{g}\left(\alpha_{i-P}\right)}+\frac{d_{P-j}}{V_{g}\left(\alpha_{P-j}\right)}.$$

The anisotropy in composite materials means that the group velocity ($V_{g}$) for the different propagated Lamb modes is direction-dependent, and therefore, the geometric angles involved in Equation (Equation 14) are defined by
(15)$$\alpha_{i-P}=\tan^{-1}\left|\frac{y_{P}-y_{i}}{x_{P}-x_{i}}\right|\;\mathrm{and}\;\alpha_{P-j}=\tan^{-1}\left|\frac{y_{P}-y_{j}}{x_{P}-x_{j}}\right|.$$

Similarly, the distances $d_{i-P}$ and $d_{P}-j$ represent the distance between transmitter i and pixel P and between pixel P and receiver j respectively. They are defined by
(16)$$d_{i-p}=\sqrt{{\left(x_{P}-x_{i}\right)}^{2}+{\left(y_{P}-y_{i}\right)}^{2}}\;\mathrm{and}\;d_{p-j}=\sqrt{{\left(x_{P}-x_{j}\right)}^{2}+{\left(y_{P}-y_{j}\right)}^{2}}.$$

The pixels with the highest contrast values are the ones where the damage is located.

## 3. Transverse Isotropic Composite Plate

The CFRC composite plate sample, whose dimensions are 500 mm × 500 mm × 6.2 mm, consists of 20 plies of 0.3 mm thick. The technical data sheets indicate that the stacking sequence is ${\left[45/0/0/45/0/45/0/45/0/45\right]}_{S}$, where the axis of symmetry is along the thickness direction. As announced in the introduction, this plate was the subject of a previous study [34] in which the optimized dispersion curves and elastic constants were determined experimentally (Table 1). Data given in Table 1 show that the symmetry is transverse isotropic, which is not in accordance with the given stacking sequence. In the following, we will consider the data given in Table 1 with a transverse isotropic symmetry. An impact was also made at the centre of this plate by means of a hemispherical impactor, thus injecting an energy of 40 joules. In practice, impacts on composite structures occur when the structure collides or is struck by an external object. Very common examples of impacts are dropping of tools during the manufacturing, preparation or storage of the structure, or during use in service, e.g., for aeronautical components, bird strikes, hail, etc. These impacts create damage that is hardly visible on the outer surfaces but which progresses through the thickness of the structure leading to local degradation of the mechanical properties around the impact location [41,42]. The problem of locating damage in this case is to determine a damaged area surrounding the impact site. An ultrasonic C-scan was therefore performed around the impacted area to estimate the extent of the area to be located.

### 3.1. Acquisitions at the Damaged State

Generation and reception of Lamb waves in different directions were performed with the help of an acquisition system whose communication with the data processing computer is wireless, as can be seen in Figure 5. This system commercially called “Geronimo” and has been developed hand-by-hand by both Gustave Eiffel University and CEA-List Laboratory. Each “Geronimo” node offers eight multiplexed channels for emission as well as for reception of acoustic waves. When emitted, acoustic waves granularity is about 10 MB/S and, when received, the maximum sampling frequency is of 2 MB/s on 16 bits for the 8 channels in parallel. In the present work, only one “Geronimo” node is exploited. However, in a network of such nodes, they can be used wirelessly and independently from each other with a local energy source (solar cells + battery, for example). A major remarkable force of each “Geronimo” node resides in its ability to be absolute-time synchronized up to some nanoseconds using GPS /PPS techniques. In other words: each node is able to send and receive acoustic waves in phase at the same moment accurately. For this purpose, each node includes the PEGASE mother-board who ensures the time-synchronization ability [43]. Eight PZT sensors (20 mm in diameter and 0.52 mm thick) are glued to the upper free surface (Z = 0 mm) of the plate. The coordinates of the sensors are given in Table 2. The excitation signal is a three-cycle Toneburst of amplitude 20 Vpp with an excitation center frequency of 150 kHz associated with a Hanning window. Its narrow band frequency content concentrates the energy around the central excitation frequency of the sensors, thus reducing the natural effects of Lamb wave dispersion.

The scan is carried out in such a way that each sensor transmits in turn when the other sensors receive, thus leading to 56 signals. The latter are acquired over a period of 500 µs at a sampling rate of 2 MHz, which is more than 10 times greater than the excitation frequency.

### 3.2. CIVA Modelling for the Creation of the Baseline State

With the help of the elastic constants of the composite plate, dispersion curves corresponding to group velocities of fundamental Lamb modes were determined. According to the scan configuration used, numerical simulations were performed in order to determine the time signals corresponding to the transmitter–receiver pairs. The reference state was thus numerically determined using the following statements:The geometry and mechanical properties of the plate (elasticity tensor, density). The plate is considered as homogeneous.The positions and size of the PZTs and the excitation signal are taken to be exactly the same as the experimental configuration.The duration of signals, i.e., 500 µs, and the sampling frequency are identical to those used in experiments.

### 3.3. Results and Discussion

The composite plate is anisotropic along the Z axis and isotropic in the (XY) plane. The speed of the A0 and S0 modes can therefore be considered as constant, where $V_{A0}\approx 1650$ m/s and $V_{S0}\approx 5700$ m/s around the 150 kHz excitation frequency. The time of flight (TOF) is determined based on these two speeds. The application of the delay-and-sum algorithm requires 56 subtractions between signals obtained experimentally at the damaged state and those obtained using the numerical simulations for each pair of sensors. Envelopes of the obtained residual signals are then determined, where an example of normalized signals taken at the reference (in blue) and damaged (in red) states is shown in Figure 6. The latter presents the residual signals for an actuator–sensor path going through the impacted zone (actuator 2—sensor 6) and for a path that does not cross it (actuator 3—sensor 5).

Figure 7 presents a through-mode C-scan performed with a Yaskawa HC20DT scanning robot. Focused piezocomposites transducers (Dasel TM) with a diameter of 28 mm are excited using the Airscope-Dasel ultrasound system. The acquisition parameters are set to obtain a maximum signal-to-noise ratio and an amplitude peak at 80% full screen height (FSH) in the least attenuating areas of the specimen. Detected damage area is circled in black and the PZTs are marked in white to be clearly observed, as can seen in Figure 8b.

Figure 8a presents the existing damage within the inspection area, obtained with a reconstruction based on the DAS algorithm assuming the A0 mode. Results show that the localization of the defect zone is close to the one given by the ultrasonic c-scan, where the damage localization error is found to be 7 mm. This result shows that the application of the DAS method based on simulated baseline signals gives an acceptable detection and localization results, even if an improvement is still necessary. This can be achieved by taking into account the attenuation in order to limit the difference between the amplitudes of the simulated and experimental signals as well as their time shift [44].

## 4. Curved Wind Turbine Blade Sample

The second sample studied is a piece of a wind turbine blade of dimensions 600 mm × 500 mm × 45 mm. It is a sandwich structure (see Figure 9), consisting of the assembly, where two glass fiber composite skins are put on both sides of a foam core. Within such a structure, different types of damage can be present such as disbonds between the skin and the core or between the laminates of the skins, cracks in the skins and more general damage can be created by impacts, as can be found in [45]. The high thickness of the sandwich structure and the porous foam attenuation properties cause Lamb waves generated from the outer face to have little chance to propagate through the foam. The conducted experiments have shown that Lamb waves are not on the inner face of the wind turbine blade, even at high excitation amplitudes. For this reason, the present study is limited to the use of Lamb waves to search for defects on the outer face of the wind turbine blade sample. The sandwich structure was impacted on the outer face away from its center. Contrary to numerous similar studies on composite structures found in the literature, in our case, we do not have any information on the draping and the material symmetry. Under these conditions, it is difficult to establish a 3D model representative of the experiment, as performed above. Furthermore, the structure has a curvature which should be taken into account. The radius of curvature of the sample studied, which is considered as a cylinder, is ≈1 m. In order to better characterize the structure, a second array of sensors is placed away from the impacted area to acquire intermediate signals. The latter are then translated according to the method described in Section 2.2 to obtain the reconstructed baseline signals.

### 4.1. Characterization of the Wind Turbine Sample

The properties of the wind turbine sample in terms of symmetry of the composite skins and their elastic constants are unknown. In the absence of such information, Lamb waves are generated in different directions with the help of a network of 16 sensors that form a circle whose center is occupied by a transmitter transducer. Group velocities of the two fundamental modes A0 and S0 are therefore determined around the excitation frequency of 75 kHz. For each propagation direction, the direct modes (A0, S0) can be differentiated by their different slope characteristics with the help of the time-frequency representation [46]. A continuous wavelet transform is performed on the collected signals, and the envelopes of the wavelet coefficients corresponding to the excitation frequency (75 kHz) are extracted to determine the different times of flight. With the help of the latter, group velocities are calculated by taking into account the radius of curvature of the sample studied, which is considered as a cylinder. Interpolation of the experimental data is then carried out to obtain the evolution of the group velocities with a step of 1°. The curves presented in Figure 10 represent the slowness of the S0 and A0 modes in the structure. It can be seen that the group velocity is maximum in the 0° direction and minimum along 90°. The symmetry of the composite studied can therefore be assumed as unidirectional, where the glass fibers are oriented in the 0°.

### 4.2. Measurements and Reconstruction of Baseline Signals

Experiments have shown that Lamb waves are highly attenuated when they propagate in the wind turbine blade. Therefore, signals are amplified to improve the transmission of the generated waves especially when they cross the damaged zones. The excitation signal is a three-cycle Toneburst delivered with a high amplitude of 400 V to compensate for losses related to irregularities as well as roughness of the glass fiber composite. The excitation frequency was around 75 kHz, the sampling frequency used for the acquisitions was 2 MHz and the acquisition time was 2.5 ms. A circular network of 400 mm diameter is positioned in the center of the upper composite skin of the structure. The network contains 16 PZT sensors of 20 mm diameter and 0.52 mm thickness. The coordinates of the sensors, which are uniformly arranged, are given in Table 3. Thus, each of the different 16 sensors can generate guided waves which will be received by the 15 other sensors.

Baseline signals were first obtained from the intact state in order to compare them with those obtained using the experimental method described in Section 2.2. Reconstructed baseline signals are obtained with the help of a second network of sensors deployed in the configuration shown in Figure 11. The latter is a reduced image of the main network having the same center. Precautions are taken to ensure that this array is located in a zone of the inspection area that does not contain damages.

The scan procedure performed in Figure 12 is the same as the one performed with the main network, which includes the damaged area. This results in 240 intermediate signals. Each signal is then processed according to Equation (Equation 12). The distance *r* in Equation (Equation 12) is taken as the difference between the path length in the main network and that of the correspondent in the secondary network. The term corresponding to the product of the exponentials is considered to be zero.

### 4.3. Results and Discussion

Results related to images obtained by considering baseline signals taken at the initial intact state (before the creation of damage) are presented in Figure 13. The difference in the time signals recorded at intact and damaged states does not seem to be important, even for signals which pass through the damaged zone, as is the case in Figure 13a. According to the literature, for damage types corresponding to delamination and/or “fiber/matrix” breakage which are created during the impact of composites, there would be no particular reflection coming from the damaged area. Instead, one should expect a small change in amplitude of the detected signals, such as the one presented in the same figure. This makes the detection of small impacts more complicated than cracks or through thickness holes as explained in references [24,47]. In our case, the small impact has been successfully detected using the delay-and-sum method (see Figure 13b). However, for the abovementioned reasons (see also the damage in Figure 11), the contrast in the image between the healthy and damaged areas is not as strong as the one corresponding to an important acoustic impedance mismatch such as a through thickness hole. In order to explore the performances of the experimental method described in Section 2.2, the damage area is detected based on the same network of sensors where the baseline state is constructed using a secondary network. Figure 14a shows signals obtained for the same ultrasonic path as in Figure 13 (actuator 10–sensor 16), where a good overlap between the two phases can be observed for the first mode identified as S0 mode. However, this is not the case for the other modes where we can observe clear differences in terms of amplitude and phase. The ultrasonic velocities corresponding to the generated modes are 905 m/s $\leq V_{A0}\leq 1852$ m/s and 1785 m/s $\leq V_{S0}\leq 2630$ m/s. Nevertheless, the image of the damaged area presented in Figure 14b, in which we used the same scaling dynamic (between 0 and 1), is similar to that obtained earlier on the same sample. The contrast obtained on the image is comparable to that presented in Figure 13b in accordance with the type of damage created by the impact. It is important to note that the reconstruction of signals is performed on the basis of the speed of S0 mode; the time of arrival corresponding to the A0 mode is therefore overestimated. In addition, the existence of an amplitude discrepancy is mainly due to the fact that inhomogeneities within the structure (which is irregular) are not taken into account in the model. We believe that these two points contribute to the existence of artifacts in the image reconstructed with the help of the delay-and-sum method. Nevertheless, the damage localization error corresponding to the presented results in Figure 13b and Figure 14b are found to be 4 mm and 7 mm, respectively. We believe that these results can be improved, but consider that for such a heterogeneous material, the results obtained present a good localization of the damaged area.

## 5. Anisotropic Carbon Fiber Composite Plate

The third sample that was characterized is a composite plate (600 mm × 600 mm × 3.5 mm) for which information related to draping, symmetry and consequently elastic properties are not available. The present study aims to detect and locate the damage created by a prior impact on the aforementioned plate for which we do not have data corresponding to the healthy or pristine state. Furthermore, there is no quantitative information on the performed impact except its position, which is at the center of the plate. In the following, group velocities of the generated Lamb waves are first determined based on scanning laser measurements. Then, intermediate signals are generated and detected with the help of a secondary network of PZT sensors before processing these signals using the experimental method described in Section 2.2.

### 5.1. Characterisation of the Composite Symmetry

In order to determine the symmetry of the composite plate, Lamb waves are generated in different directions using 150 kHz Gaussian burst signals. Scanning laser measurements are performed to detect the out-of-plane displacement according to the different propagation directions that surround a transmitting transducer (area = 30 cm × 30 cm) as shown in Figure 15a,b. In view of the excitation frequency, the wave packets are identified as S0, SH0 and A0, respectively. The propagation of the wave packets is observed before the mode S0 (the fastest mode) is reflected on the edge of the plate or by the damaged area, which corresponds to a propagation time of 65 µs, as shown in Figure 15b. The latter also shows the diamond-shaped distribution of the Lamb wave velocities symmetrically between 0° and 360°. Two main directions of propagation are therefore identified, namely 0° and 90°, along which the velocity of the different modes is maximum, where $V_{S0}=5712$ m/s, $V_{A0}=1904$ m/s and $V_{SH0}=3808$ m/s.

By interpolating the experimental data recorded in the different propagation directions, a complete representation of the slowness curves for the three fundamental Lamb waves is obtained, as presented in Figure 15c. According to literature [48], and based on the experimental results, the symmetry of the composite plate was identified as quasi-isotropic-transverse. The latter corresponds to an orthotropic symmetry invariant by a rotation of 90° around a single axis of the orthotropic reference frame and is characterized by six independent elasticity moduli [3,4,49].

### 5.2. Damaged State and Baseline Signals

In order to locate the damage within the anisotropic composite plate, 12 evenly distributed PZT sensors are used to cover a square inspection area with an edge equal to 450 mm, as shown in Figure 16. Lamb waves are generated around 150 kHz with an amplitude corresponding to 30 Vpp. The impact damage is situated in the center of the plate. The scan is carried out in the same way as described earlier to give 132 signals, which are acquired over a period of 500 µs at a sampling rate of 2 MHz.

In order to obtain baseline signals, a second network is used to cover a square inspection area with an edge equal to 100 mm away from the impact. A total of 132 intermediate signals are then obtained and modified according to Equation (Equation 12), with the same assumptions used earlier, to obtain baseline signals.

### 5.3. Results and Discussion

Results related to the baseline signal and the damaged state signal are presented in Figure 17. In the latter, the reconstructed baseline signals (in red) and the damaged signals (in blue) are presented when the ultrasonic path does not cross the damaged area (actuator 1–sensor 3) (Figure 17a) and when it goes through it (actuator 6–sensor 12) (Figure 17b). Away from the defect area, a strong resemblance can be observed between the signals originating from the path which connects sensors 1 and 3. For the path between the sensors 6 and 12, it can be seen that the presence of damage has a visible effect that can be observed on almost all the recorded signals with equivalent paths in terms of amplitude and time shift.

Localization results obtained based on the delay-and-sum algorithm are presented in Figure 18. The latter considers different incident waves according to the modes generated, namely the A0, SH0 and S0 modes, respectively. Results show a good agreement with the real impact location (red circle in Figure 18a–c) when the incident Lamb wave is taken as A0 (Figure 18c) and S0 (Figure 18a). The damage localization errors are found to be 3 mm and 4 mm when the incident Lamb wave is taken as A0 and S0, respectively. However, results related to the case when the incident Lamb wave is SH0 (Figure 18b) does not allow obtaining a successful localization. The significant change in the waveform shows that the wave packet related to the SH0 mode, which has been seriously affected by the interaction with damage, suggests the existence of a mode conversion by damage in the plate. This hypothesis is supported by the presence of the time shift between the wave packets, such as the one corresponding to the path 6–12 [50]. In general, the generated Lamb modes (A0, S0 or SH0) are expected to be either directly transmitted, reflected from an edge (of the structure or damage) or converted [49]. Some mode conversions only generate modes that share the same symmetry as the incident mode. This is the case of symmetric discontinuities with respect to the plate’s mid-plane [51]. In cross-ply laminates, when the A0 mode interacts with a delamination that is located asymmetrically across the thickness, the mode-converted S0 is observed. The mode-converted S0 is revealed to be confined within the delamination region and converts back to the A0 mode when delamination is symmetrically located across the thickness [52,53]. SH guided, however, are classified between symmetric and anti-symmetric according to their displacement profile across the plate’s thickness. The interaction of SH waves with a discontinuity can be complex depending on the discontinuity shape as well as the operating frequency–thickness product [54,55]. From the incidence of a single SH mode upon a thickness discontinuity (such as the one created by an impact), any mode can arise due to mode conversion. This makes the estimation of the conversion that took place very difficult in particular in an anisotropic material such as the composite plate studied. In the absence of a valuable information about the existing conversion, different combinations can be applied leading to a multi-modal reconstruction according to the involved propagating modes. For example, in the delay-and-sum algorithm, mode conversion can be taken into account by differentiating the speed of the incident mode between the transmitter and the pixel and that of the converted mode between the pixel and the receiver in Equation (Equation 14).

## 6. Conclusions and Perspectives

This paper presented different baseline-free methodologies to locate damage within composite structures using guided waves. The development of numerical simulations has allowed obtaining baseline signals corresponding to a transverse isotropic composite plate with known elastic properties. The delay-and-sum algorithm applied on experimental and simulated signals has allowed for detection and reasonable localization of the impact damage despite the simulation time, which can be quite long depending on the case studied. When the elasticity of the material is not known (or is difficult to obtain), this paper proposes an experimental method in which Lamb wave velocities are determined in different directions with the help of a second PZT network more distant from the edge of the sample than the main network. Baseline signals were then obtained with the help of a propagation model which can be adapted to anisotropic materials. In the case of a curved wind turbine blade sample (which was revealed to be unidirectional), the relevance of this new approach has been tested by comparing the results obtained with those from measurements with a reference state. We have proven that reasonable localization errors can be found with the help of the proposed reconstruction method of the baseline signals which did not exceed 7 mm despite the heterogeneity and the geometrical irregularities of the sample. The method was then applied on an impacted composite whose symmetry is unknown. Based on the experimental measurements, the slowness curves corresponding to the different guided waves helped to conclude that the composite symmetry is quasi-isotropic transverse. The localization of the defect was found to be 3 mm and 4 mm by considering the modes A0 and S0, respectively. This work has shown that there is a real interest in using different propagation modes in order to optimize the localization of impacts on different types of composites. Indeed, contrary to the modes A0 and S0, the localization of the defect area was not possible with the help of the SH0 mode, most probably due to mode conversion, as discussed in the paper. This opens the perspective of developing a multi-modal imaging method that takes into account the different propagation modes as well as the possible mode conversions that may happen within the material studied. Future work will give attention to developing an optimization of the sensors’ positions (symmetric or evenly spaced), which has been shown to be an important parameter in damage localization accuracy, in order to minimize the number of transducers used. The optimized network will be used to image stiffened composite panels taken at rest and noisy environmental conditions such as vibration.

## Figures and Tables

**Figure 1 sensors-23-04368-f001:**
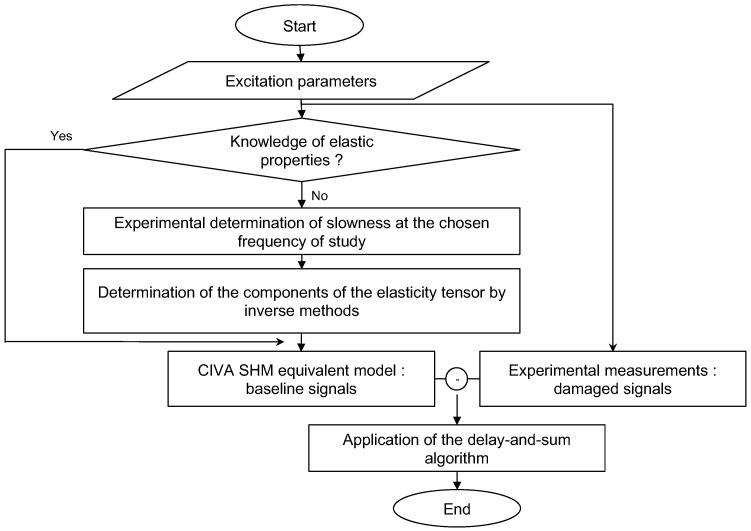
Method 1 based on the construction of baseline signals by numerical simulations.

**Figure 2 sensors-23-04368-f002:**
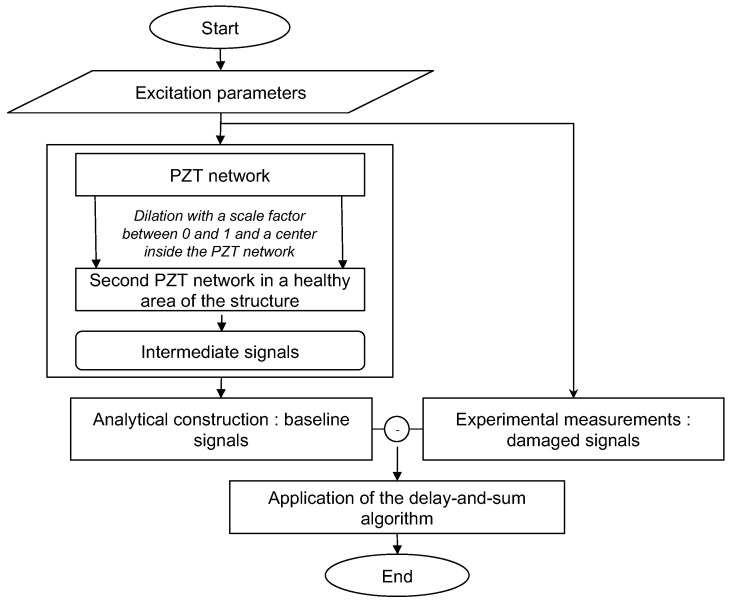
Method 2 based on an analytic construction of baseline signals.

**Figure 3 sensors-23-04368-f003:**
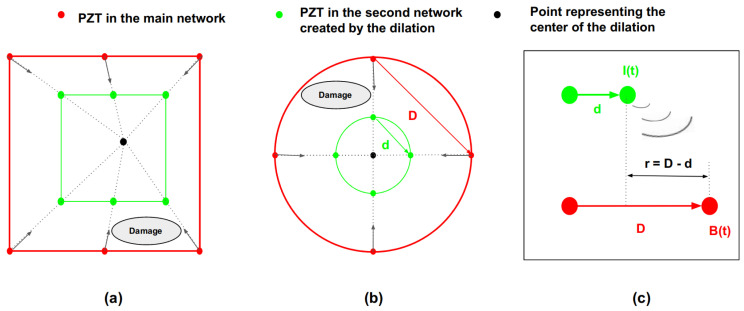
Method 2 baseline signals creation process: (**a**) dilation of a square distribution of six PZT. (**b**) dilation of a circular distribution of four PZT. (**c**) From intermediate signal to reconstructed baseline signal.

**Figure 4 sensors-23-04368-f004:**
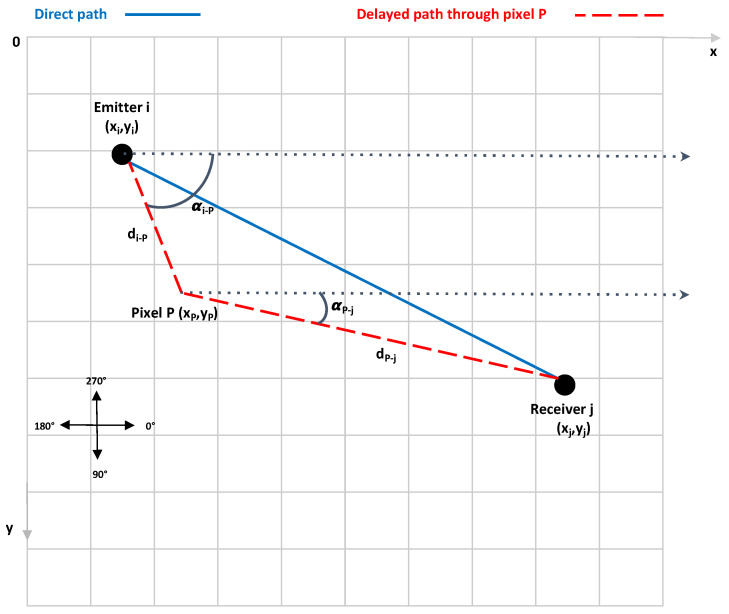
Delay-and-sum algorithm used in the two methods.

**Figure 5 sensors-23-04368-f005:**
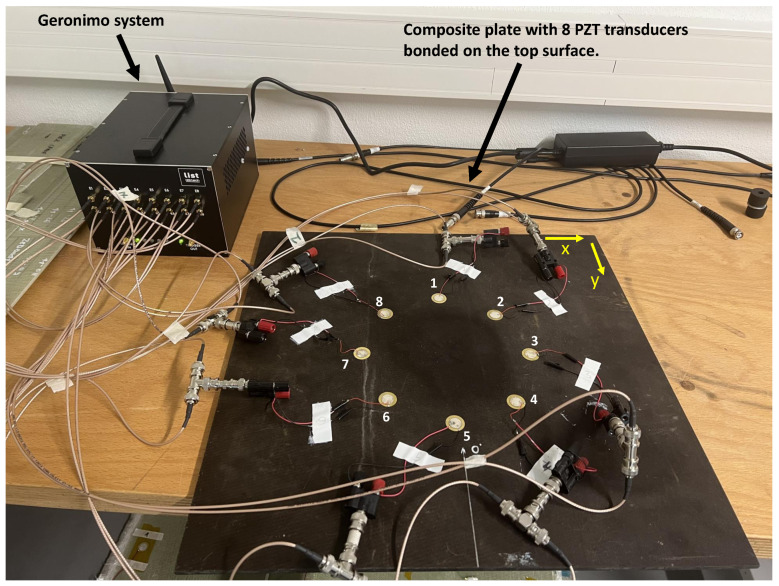
Experimental setup for damaged signals acquisition.

**Figure 6 sensors-23-04368-f006:**
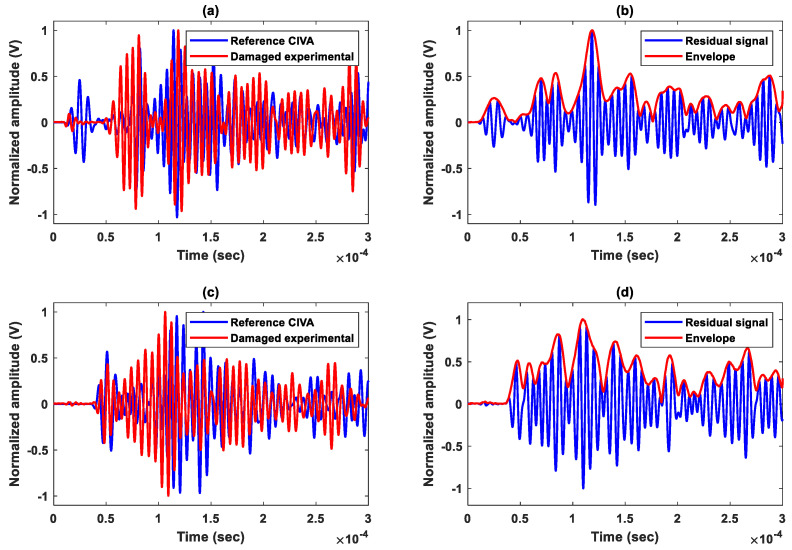
Baseline, damaged, residual and envelope of signals of two paths with one going through damage area and the other not: (**a**,**b**) path (actuator 2—sensor 6); (**c**,**d**) path (actuator 3—sensor 5).

**Figure 7 sensors-23-04368-f007:**
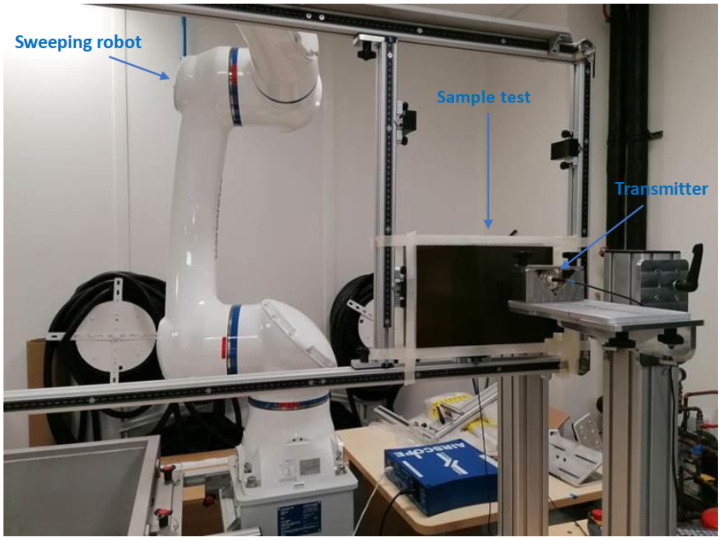
Experimental setup of ultrasonic C-scan.

**Figure 8 sensors-23-04368-f008:**
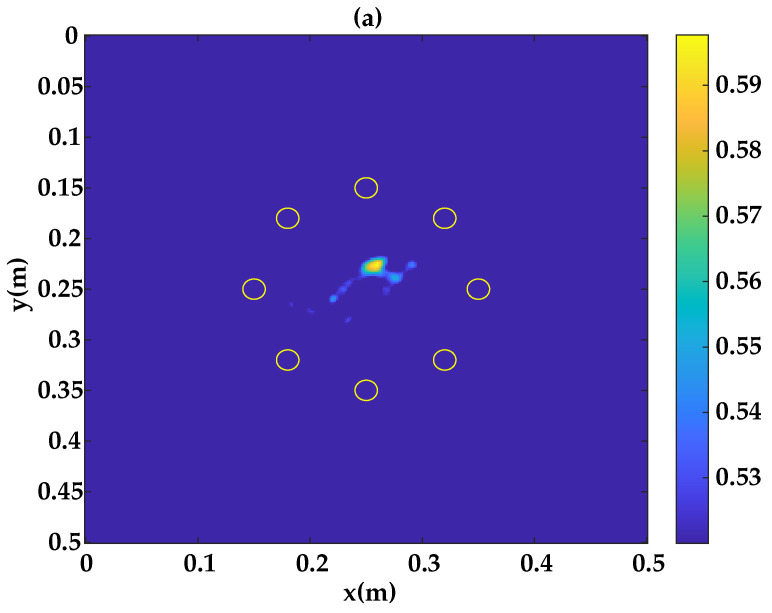
Damage localization results: (**a**) Method 1. (**b**) Ultrasonic C-scan.

**Figure 9 sensors-23-04368-f009:**
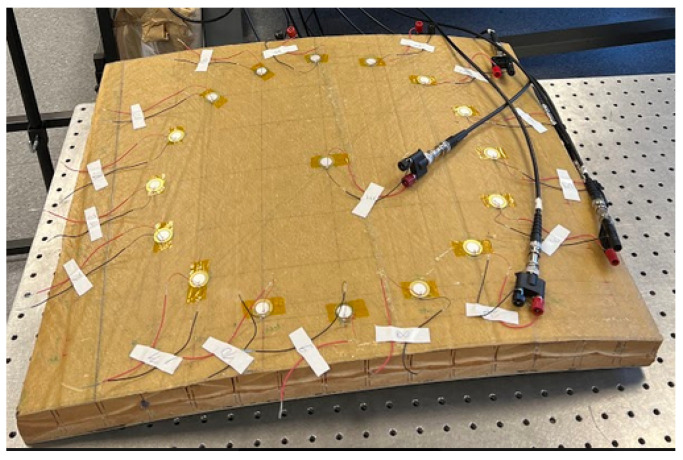
Curved wind turbine blade sample.

**Figure 10 sensors-23-04368-f010:**
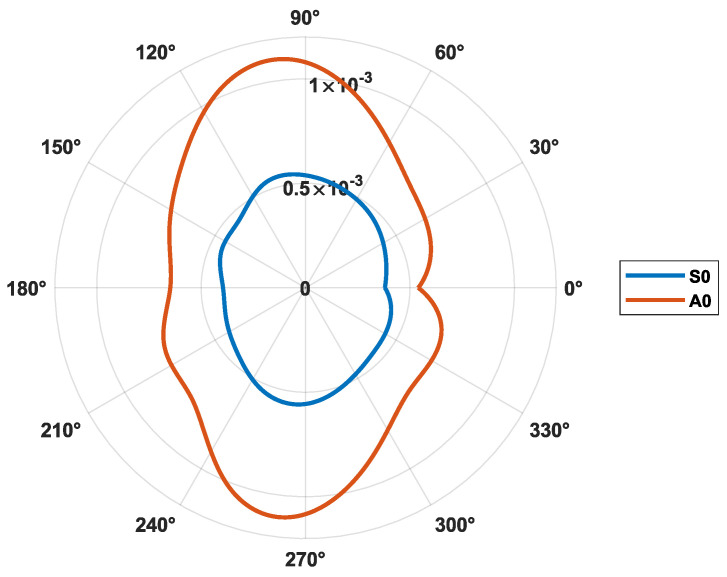
Slowness of S0 an A0 at the 75 kHz excitation frequency.

**Figure 11 sensors-23-04368-f011:**
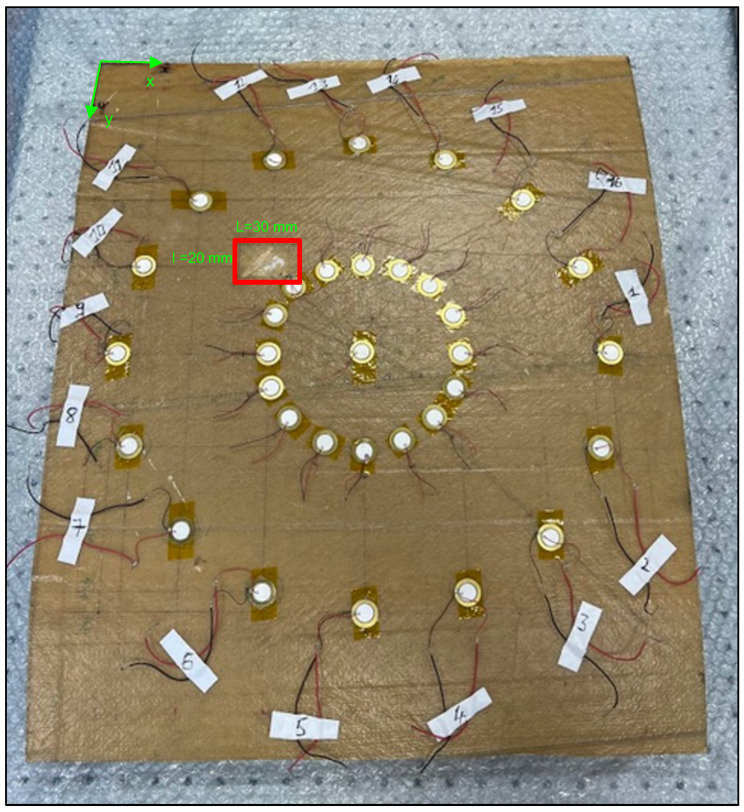
Dilation of the main PZT network for the acquisition of intermediate signals and the damaged area inside the red rectangle.

**Figure 12 sensors-23-04368-f012:**
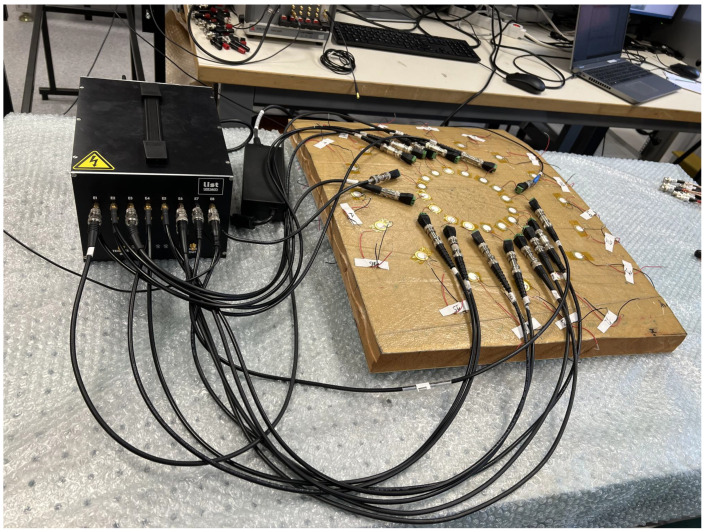
Acquisition of the intermediate signals.

**Figure 13 sensors-23-04368-f013:**
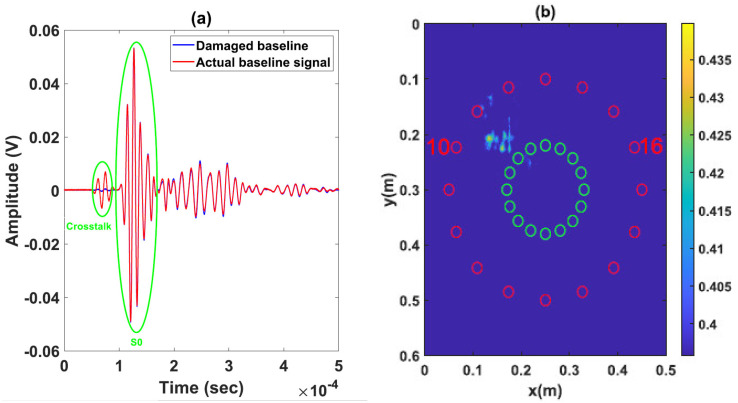
Damage localization results with delay-and-sum using actual baseline signals: (**a**) Actual baseline and damaged signals for one path (actuator 10–sensor 16) going through damage area. (**b**) Reconstruction with S0 mode.

**Figure 14 sensors-23-04368-f014:**
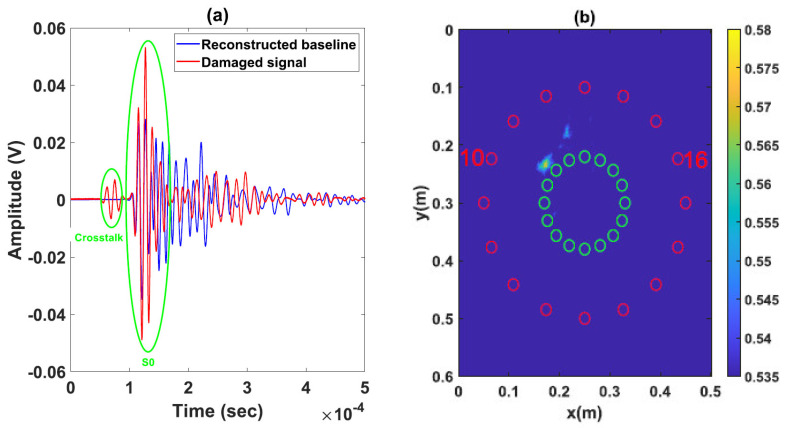
Damage localization results with method 2 using reconstructed baseline signals: (**a**) Reconstructed baseline and damaged signals for one path (actuator 10–sensor 16) going through damage area. (**b**) Reconstruction with S0 mode.

**Figure 15 sensors-23-04368-f015:**
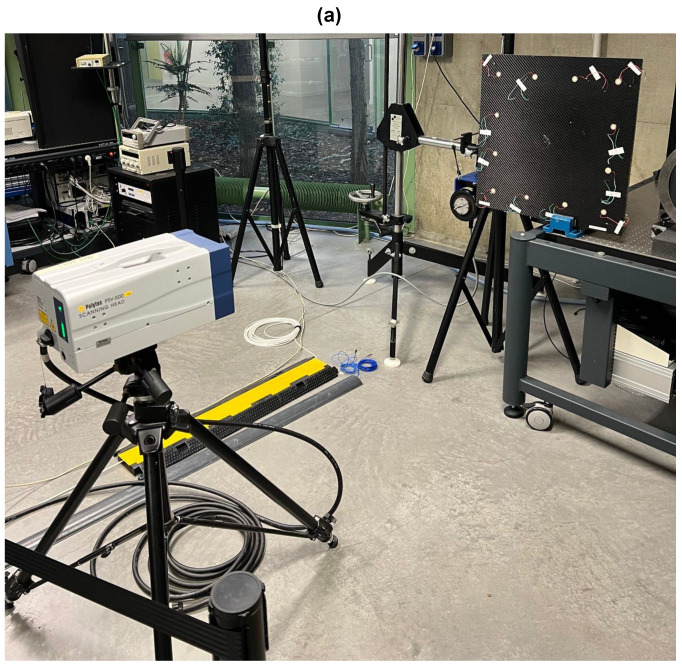
Characterization of anisotropy: (**a**) Schematic for measuring group velocities of A0, S0 and SH0 Lamb wave modes using laser. (**b**) Lamb Waves propagation after 65 µs. (**c**) Slowness of S0, SH0, A0 at the 150 KHz excitation frequency.

**Figure 16 sensors-23-04368-f016:**
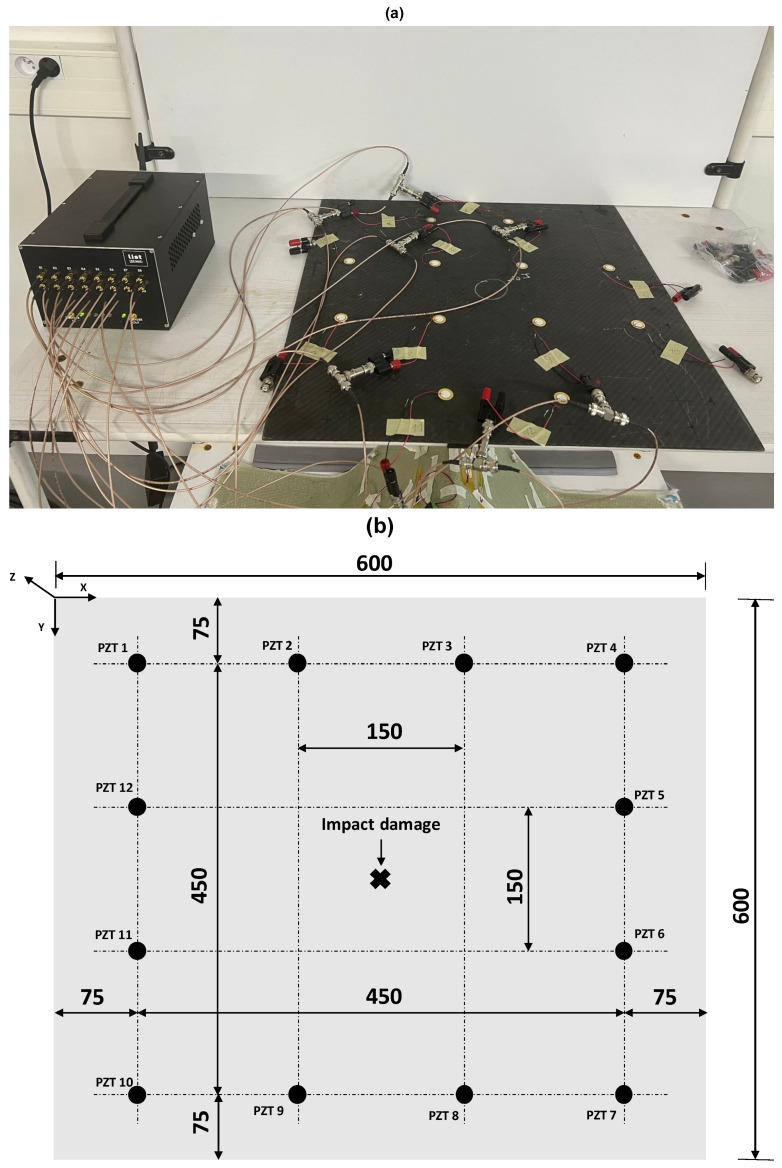
Schematic of the composite plate: (**a**) Experimental setup. (**b**) PZT locations and the impact damage position.

**Figure 17 sensors-23-04368-f017:**
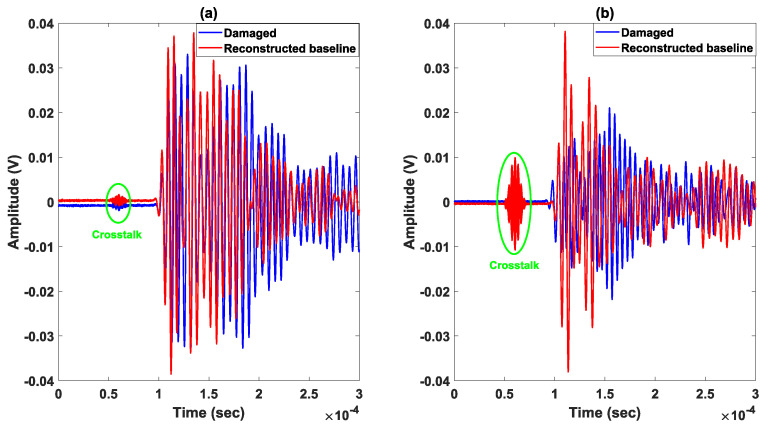
Method 2 signals of two paths with one (**a**) not going through damage area and the other (**b**) going through it: (**a**) Path (actuator 1–sensor 3). (**b**) Path (actuator 6–sensor 12).

**Figure 18 sensors-23-04368-f018:**
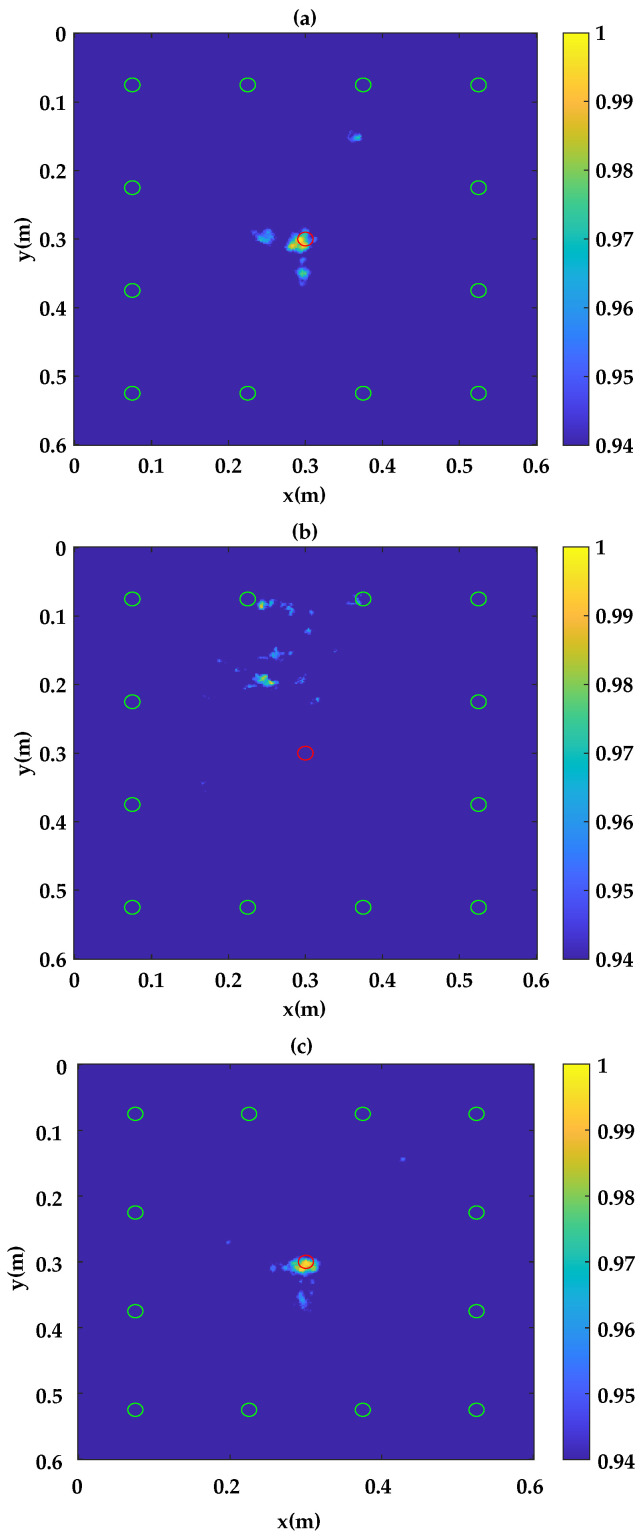
Damage localization results with method 2: (**a**) Reconstruction with S0. (**b**) Reconstruction with SH0. (**c**) Reconstruction with A0.

**Table 1 sensors-23-04368-t001:** Homogenized elastic properties of the composite plate.

$\rho (g/{\mathrm{cm}}^{3})$	$C_{11}\left(\mathrm{GPa}\right)$	$C_{22}\left(\mathrm{GPa}\right)$	$C_{33}\left(\mathrm{GPa}\right)$	$C_{12}\left(\mathrm{GPa}\right)$	$C_{13}\left(\mathrm{GPa}\right)$	$C_{23}\left(\mathrm{GPa}\right)$	$C_{44}\left(\mathrm{GPa}\right)$	$C_{55}\left(\mathrm{GPa}\right)$	$C_{66}\left(\mathrm{GPa}\right)$
1.55 ± 0.05	52.4 ± 2%	52.4 ± 2%	12.6 ± 2%	17.8 ± 2%	3.1 ± 2%	3.1 ± 2%	3.7 ± 2%	3.7 ± 2%	17.3 ± 2%

**Table 2 sensors-23-04368-t002:** Coordinates of the PZT.

Positions	PZT1	PZT2	PZT3	PZT4	PZT5	PZT6	PZT7	PZT8
X (mm)	250	320	350	320	250	180	150	180
Y (mm)	150	180	250	320	350	320	250	180

**Table 3 sensors-23-04368-t003:** Coordinates of the PZT.

PZT	X(mm)	Y(mm)
1	450	300
2	434.8	376.5
3	391.4	441.4
4	326.5	484.8
5	250	500
6	173.5	484.8
7	108.6	441.4
8	65.2	376.5
9	500	300
10	65.2	223.5
11	108.6	158.6
12	173.5	115.2
13	250	100
14	326.5	115.2
15	391.4	158.6
16	434.8	223.5

## Data Availability

Not applicable.

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
