# Peer review of "Damage Localization on Composite Structures Based on the Delay-and-Sum Algorithm Using Simulation and Experimental Methods"

_sensors, 2023, doi:10.3390/s23094368_

Round 1

Reviewer 1 Report

Manuscript ID: sensors-2305899

Title: Damage localization on composite structures based on the delay-and-sum algorithm using simulation and experimental methods

This study focuses on the Lamb wave localization of damages within different anisotropic composite structures based on the use of piezoelectric transducer networks. This paper is well organized, however several points presented below need improvement. My final recommendation is to accept after minor revisions.

1. Please check for typos.

2. The quality of some figures needs to be improved.

3. Some subtitles should be rewritten, such as 3.3, 4.3, 5.3 Results and discussion.

4. The objective of the work is not well presented and clear to determine the novelty of the work.

Author Response

Reviewer 1

We would like to thank the reviewer for his comments and suggestions in order to improve the quality of the paper. Our response to the raised points can be fin be below:

Comments and Suggestions for Authors

Manuscript ID: sensors-2305899

Title: Damage localization on composite structures based on the delay-and-sum algorithm using simulation and experimental methods

This study focuses on the Lamb wave localization of damages within different anisotropic composite structures based on the use of piezoelectric transducer networks. This paper is well organized, however several points presented below need improvement. My final recommendation is to accept after minor revisions.

  1. Please check for typos.

Done. Thank you.

  1. The quality of some figures needs to be improved.

The revised version contains figures of a better quality.

  1. Some subtitles should be rewritten, such as 3.3, 4.3, 5.3 Results and discussion.

Thank you for your comments. Subtitles have been reconsidered in adequacy with the revised version.

  1. The objective of the work is not well presented and clear to determine the novelty of the work.

The revised version presents the objectives of the work in a better way.

Thank you for all your valuable comments and suggestions.

Reviewer 2 Report

The authors investigated damage detection in composite structures using the delay-and-sum algorithm. In the reviewer’s opinion, the paper is interesting, and the results can ease future work in this regard. However, there are a couple of comments that need to be addressed before the paper can be considered for publication. Please see my comments below:

1-      The novelty and importance of the study have not been clearly shown. Please add more detail to indicate why this study is critical and worth investigating.

2-      Please provide units for all equations used in this study.

3-      Please provide more figures for the experimental work conducted.

4-      The English language of the paper should be revised. There are some grammatical issues, including mixing active and passive voice, spelling, and singular vs. plural, etc. A grammatical review is recommended to clean up all these issues.

Author Response

Reviewer 2

We would like to thank the reviewer for his comments and suggestions in order to improve the quality of the paper. Our response to the raised points can be fin be below:

Comments and Suggestions for Authors

The authors investigated damage detection in composite structures using the delay-and-sum algorithm. In the reviewer’s opinion, the paper is interesting, and the results can ease future work in this regard. However, there are a couple of comments that need to be addressed before the paper can be considered for publication. Please see my comments below:

1-      The novelty and importance of the study have not been clearly shown. Please add more detail to indicate why this study is critical and worth investigating.

We thank the reviewer for her/his comment. The revised version presents this point in a better way. Changes are written in blue.

2-      Please provide units for all equations used in this study.

Units have been added for each parameter as requested.

3-      Please provide more figures for the experimental work conducted.

We have added new figures for the experimental work in the revised version.

4-      The English language of the paper should be revised. There are some grammatical issues, including mixing active and passive voice, spelling, and singular vs. plural, etc. A grammatical review is recommended to clean up all these issues.

We have mts to made efforts to improve the English of this paper.

We thank you for all your comments and suggestions.

Reviewer 3 Report

I have attached file to this email

Author Response

Dear Sir or Madame,

We wish to thank you for your valuable comments and suggestions. 

Please find the answers to your questions in the attached file. 

Best regards,

Reviewer 4 Report

Comments to the Author(s)

This manuscript presents a Damage localization on composite structures based on the delay-and-sum algorithm using simulation and experimental methods. This paper contains a good effort related to SHM and NDT community because it presents interesting methods based on lamb waves for visualizing damages. In general, the manuscript is organized well. However, The authors have to improve the paper by resolving the following issues (All the answers should be included in the manuscript):

1.      Fig 3, How did you determine the location of the dilation of a square distribution of six PZT and the dilation of a circular distribution of four PZT (r-value)? In the case of multiple damages or unknown damage locations, Can this method be applied practically to deal with these cases? Please clarify.

2.      Page 7, line 192, Please rewrite the following sentence “The anisotropy in composite materials means that the group velocity of the different…” to be “The anisotropy in composite materials means that the group velocity (Vg) of the different…”

3.      Page 7, line 201, instead of writing “Dimensions of the plate are of 500 × 201

4.      500 x 6.2 mm3”, please write “Dimensions of the plate are of 500 mm × 201 mm

500 × 6.2 mm”

5.      Page 8, line 213, Please add the Fig. of the ultrasonic C-scan image of the test plate and explain it.

6.      Fig. 5, Please label the items of the experimental setup.

7.      Fig. 6, please rewrite the caption of Fig.6 to be “Figure 6. Baseline, damaged, residual, and envelope of signals of two paths with one going through the damage area and the other not: (a),(b) path (actuator 2 - sensor 6); (c),(d) path (actuator 3 - sensor 5).”

8.      Page 10, line 260, please add space between the speed value and its unit for the following: 1650m/s and 5700m/s, and mention the wave frequency value related to this speed value.

9.      Please label the items in Fig. 7(b).

10.  Fig. 10, the main PZT network has strong reflected waves from the edges compared with the Dilation PZT network (secondary network) which affect the result of damage localization. How did you deal with this issue?

11.  Fig. 12b, please add the second network to the plot with a different color.

12.  Fig.16, Because you are making a comparison between three images, they should have the same color map scale, please unify the color map scale for all three images.

Author Response

Please find our answers in the attached file. 

Thank you for your valuable comments and suggestions.

Best regards,

Reviewer 5 Report

This paper presents two methods to obtain baseline data, which are simulation and experiment. A lot of experiments have been done.

1. Section 3

Could you please give some explanation about the composite plate being anisotropic along the Z axis and isotropic in the (XY) plane? How could [45/0/0/45/0/45/0/45/0/45]s be transverse isotropic?

2. Section 4

In section 4, the authors mentioned that the structure has a curvature which should be taken into account. But I didn’t find what work has been done by the authors regarding the curvature of structures?

3. The font in figures is suggested to be enlarged, especially fig.9, 11,12, 15

Author Response

Dear Sir or Madame,

The answers are in the attached file. 

Thank you for your time and valuable comments and suggestions. 

Round 2

Reviewer 2 Report

The authors revised the paper adequately. Accordingly, it can be accepted.

Reviewer 3 Report

The corrections were done